# Radiation-Induced Breast Angiosarcoma—A Single-Institution Experience

**DOI:** 10.3390/diagnostics14202326

**Published:** 2024-10-18

**Authors:** Marko Buta, Nada Santrac, Milan Zegarac, Merima Goran, Nikola Jeftic, Nevena Savkovic, Jovan Raketic, Saska Pavlovic, Ognjen Zivkovic, Aleksandar Rankovic, Ivan Markovic

**Affiliations:** 1School of Medicine, University of Belgrade, Dr Subotica 8, 11000 Belgrade, Serbia; milan_zegarac@yahoo.com (M.Z.); merimaoruci@hotmail.com (M.G.); ivanmarkovic66@yahoo.com (I.M.); 2Surgical Oncology Clinic, Institute for Oncology and Radiology of Serbia, Pasterova 14, 11000 Belgrade, Serbia; jefta20@gmail.com (N.J.); nevena5gm@gmail.com (N.S.); jovanraketic96@gmail.com (J.R.); saskapavlovic23@gmail.com (S.P.); 3Department of Pathology, Institute for Oncology and Radiology of Serbia, Pasterova 14, 11000 Belgrade, Serbia; drognjenzivkovic@yahoo.com; 4Radiation Oncology Clinic, Institute for Oncology and Radiology of Serbia, Pasterova 14, 11000 Belgrade, Serbia; alekranko@gmail.com

**Keywords:** angiosarcoma, breast cancer, irradiation/radiotherapy, radiation induced, treatment, prognosis

## Abstract

**Introduction**: Radiation-induced breast angiosarcoma (RIBAS) is a rare adverse event associated with postoperative breast irradiation. The data from the literature indicate that RIBAS occurs in less than 0.3% of patients treated with adjuvant radiotherapy for breast cancer. Given the rarity, diverse clinical presentation, poor prognosis, and lack of consensus on the management, this study aimed to present experiences of our specialized cancer center with RIBAS, in terms of the incidence, presentation, management, and outcomes. **Methods**: We reviewed the medical records of 10,834 breast cancer patients treated at the Institute for Oncology and Radiology of Serbia between January 2013 and June 2024 to detect patients that had breast-conserving surgery, followed by postoperative irradiation, and developed angiosarcoma in the irradiated area at least 3 years after radiotherapy, without distant metastases. The incidence, latency period, management, and treatment outcomes were analyzed. **Results**: A total of nine female patients with RIBAS were identified and included in this study. The median age at RIBAS diagnosis was 64 years (range: 36–68), with a median latency of 64 months (95% CI > 57) from irradiation to diagnosis. The mean tumor size was 55 mm (SD 32.78). Patients were followed for a median of 30 months (range: 7–40) after initial RIBAS surgery. Local recurrence occurred in seven patients (77.8%), with five undergoing re-do surgery with curative intent. Three patients developed distant metastases during follow-up. The median overall survival (OS) was 31 months (95% CI > 30), with a 3-year survival rate of 15.2% (95% CI 2.5–91.6%). The median local recurrence-free interval was 10 months (95% CI > 3). Median OS after RIBAS local recurrence and after breast cancer treatment was 17 months (95% CI > 15) and 108 months (95% CI > 88), respectively. **Conclusions**: RIBAS is a rare but increasingly prevalent adverse event associated with BC irradiation, marked by an aggressive disease course and high relapse rates. Awareness, prompt diagnosis, and a radical surgical approach with wide clear margins are critical for improving patients’ outcomes.

## 1. Introduction

Radiation-induced breast angiosarcoma (RIBAS) is a rare adverse event associated with postoperative breast radiotherapy (RT) that is a standard of care for early breast cancer treatment [1,2]. The data from the literature indicate that RIBAS occurs in less than 0.3% of patients treated with adjuvant radiotherapy for breast cancer [3]. These aggressive tumors develop from the blood vessels in the breast tissue and can manifest with symptoms like rapid growth, pain, and skin discoloration. Diagnosing and treating RIBAS can be challenging, making it essential to understand the risk factors, clinical presentation, and treatment options.

In 1948, Cahan et al. first outlined criteria for post-bone-irradiation sarcoma, which were later adapted for RIBAS by Arlen et al. in 1971. These criteria include the presence of a primary malignant tumor with different histology from the radio-induced sarcoma, the location of the angiosarcoma within the irradiation field, a minimum latency period of 3 years between the two malignancies, and the second malignancy being a sarcoma [4,5]. These criteria were also used in our study.

Given the rarity, diverse clinical presentation, poor prognosis, and lack of consensus on the management, this research aimed to present experiences of our specialized cancer center with RIBAS, in terms of the incidence, presentation, management, and outcomes.

## 2. Materials and Methods

We retrospectively reviewed medical records of 10,834 breast cancer (BC) patients treated at the Surgical Oncology Clinic, Institute for Oncology and Radiology of Serbia (IORS), between January 2013 and June 2024. Study inclusion criteria were as follows: (1) initial diagnosis of breast cancer, (2) treatment with breast-conserving surgery, (3) postoperative RT applied, (4) histological verification of angiosarcoma in the irradiated area at least 3 years after RT, and (5) absence of metastatic disease at the time of angiosarcoma diagnosis. All patients included in this study received complete BC treatment at IORS.

Data concerning patients’ age, primary tumor characteristics (type, TNM staging), surgical interventions (breast/axillary), RT protocols (dose, boost), and other treatments (endocrine therapy or chemotherapy) were collected from medical records and pathology reports and subjected to analysis.

Initial analysis was focused on the incidence, latency period (defined as the time from BC RT to RIBAS diagnosis), and treatment of RIBAS.

Further analysis was performed in terms of outcomes after RIBAS diagnosis and treatment, including overall survival (OS) and local recurrence-free interval (LRFI). OS was defined as the time from initial RIBAS treatment to death or the last follow-up. LRFI was defined as the time from initial RIBAS surgery to first pathohistologically proven local recurrence. Additionally, overall survival after initial BC surgery (OS-BC) and overall survival after RIBAS local recurrence (OS-LR) were also studied.

## 3. Results

Out of 10,834 patients treated for BC at the IORS between January 2013 and June 2024, 7647 received postoperative RT based on indications and guidelines. A total of nine female patients met the inclusion criteria for this study. 

The characteristics and treatment approach for the initial BC are shown in Table 1. The median age at BC diagnosis was 53 (range: 31–64) years. All nine patients in our study had hormone receptor-positive, grade 2 or, less frequently, grade 1 tumors. The disease stage was defined based on the AJCC classification that was valid at the time of BC diagnosis. Patients were treated based on the national guidelines that align with ESMO recommendations. All patients had early breast cancer. Seven of them received breast-conserving surgery with axillary lymph node dissection, while two remaining patients underwent breast-conserving surgery with sentinel lymph node biopsy. All nine patients underwent postoperative breast RT, were treated under the same technical conditions (on linear accelerators with 6MV energies, Elekta Synergy up to 2018, and then Varian Clinac and TrueBeam machines until 2023). Radiotherapy planning was conducted with a 3d conformal radiotherapy technique for all patients. 

Figure 1, Figure 2 and Figure 3 show the RIBAS clinical presentation and pathology results. Detailed RIBAS clinical presentations and treatment strategies are summarized in Table 2 and Table 3. No patient had active BC at the time of RIBAS diagnosis, nor did any patient develop recurrent BC during the RIBAS follow-up. Median age at RIBAS diagnosis was 64 (range: 36–68) years, while the estimated median time from BC RT to RIBAS diagnosis was 64 (95% CI > 57) months. Mean tumor size at RIBAS diagnosis was 55 (SD 32.78) mm. All nine patients had localized disease and underwent total mastectomy with curative intent. Four patients needed pectoral defect reconstruction. A total of four patients received adjuvant chemotherapy, three of whom received doxorubicin as monotherapy, while the remaining patient had doxorubicin and cyclophosphamide combination (AC protocol). The patients were followed up for a median of 30 (range: 7–40) months after the initial RIBAS operation. Local recurrence was observed in seven patients, five of whom had surgery with curative intent. Three patients developed distant metastases during follow-up. 

The median OS was 31 (95% CI > 30) months with an estimated 3-year survival rate of 15.2% (95% CI 2.5–91.6%) (Figure 4A). The estimated median LRFI was 10 (95% CI > 3) months (Figure 4B). The estimated median OS-LR and OS-BC were 17 (95% CI > 15) and 108 (95% CI > 88) months, respectively (Figure 4C,D).

## 4. Discussion

Radiation-induced angiosarcoma was first reported in 1981 by Maddox and Evans [6]. With the wider adoption of breast-conserving surgery for BC in combination with postoperative RT, and modern medical treatments becoming more accessible throughout the world, the number of patients with RIBAS is expected to rise. However, the therapeutic strategies for RIBAS show considerable variability across medical institutions, reflecting an absence of standardized treatment protocols. 

Several theories have been proposed to explain the development of RIBAS. One theory suggests that photons directly damage DNA, causing breaks and leading to genomic instability and mutations in cancer-related genes. Another theory proposes that reactive oxygen species generated by radiation can further damage DNA, proteins, and lipids [7]. The minimum total dose required to induce RIBAS was reported to be 10Gy [8]. Rombouts stated in a large population-based study that none of the patients who did not receive RT for BC treatment (n = 111,754) developed angiosarcoma. On the other hand, out of 184 823 patients who received RT, 209 (1 of 1000) developed RIBAS of the breast and/or chest wall [9]. RIBAS has also been associated with specific gene mutations, such as inactivation of the tumor suppressor gene p53 and amplification of the 8q24 region containing the MYC oncogene [10]. The development of secondary angiosarcoma in chronic lymphedema following radical mastectomy for BC was first described in 1948 by Stewart and Treves, after whom the condition was later named [11]. Chronic lymphedema caused by axillary lymph node dissection and RT can contribute to the development of RIBAS by promoting tumorigenesis through vascular growth factor stimulation in the tumor microenvironment [12]. Having this in mind, it is worth noticing that in our study seven out of nine patients underwent axillary lymph node dissection. Gene amplification of FLT4 (VEGFR3) and mutation of KDR (VEGFR2) have been linked with RIBAS development [13]. Additionally, there may be a connection between the breast cancer-related tumor suppressor genes BRCA1/BRCA2 and RIBAS, although the exact mechanism is not yet fully understood [14].

According to the data from our study, RIBAS developed in 0.12% of irradiated patients, which is similar to other published studies (3, 5, 7). Several studies have reported median ages at diagnosis ranging from 58 to 72 years [5,9,15,16,17], which are in line with the 64 years in our study. The median latency period of 64 months (5.3 years) is comparable to other studies which reported median latency times ranging from 4.9 to 9.2 years [9,16,18]. RIBAS usually presents insidiously, with skin changes such as discoloration (ranging from red to purple), elevated skin, and skin thickening, which sometimes can be mistaken for benign skin lesions [19], potentially leading to delayed diagnosis at more advanced stages. Evidence of any such lesion in irradiated breasts must be taken seriously and referred for further evaluation. In addition to physical examination, radiological evaluation and biopsy are the key steps in reaching diagnosis. Although mammography, as a part of regular BC follow-up, can be of help in some patients, the large number of false-negative mammography results, even with the presence of skin changes, makes MRI the method of choice for evaluation of these patients [19,20]. Biopsy and histological examination ultimately confirm the diagnosis of RIBAS. Biopsy of the affected skin is usually sufficient for the diagnosis of most cutaneous presentations of RIBAS, while deeper lesions may require fine-needle aspiration or core-needle biopsy [21,22,23]. Early detection and prompt treatment may increase the number of patients diagnosed in the early stages of the disease, providing better local and distant disease control. The mean tumor size of 55 (SD 32.78) mm in our study is consistent with the 50 mm reported by other teams [15,18]. The tumor size was found to be a predictive factor for long-term outcomes [15,24]. Barrow reported that patients with breast sarcomas under 2 cm in size had a median OS of 80 months, compared to a mere 20 months in patients with sarcomas >5 cm [25]. Furthermore, RIBAS can frequently present as a multifocal lesion, which has also been found to determine poor prognosis in these patients [5]. Thus, obtaining negative margins in patients with large multifocal tumors can present a very challenging task even for experienced teams. 

The treatment of choice for all RIBAS patients presenting with resectable disease is total mastectomy, with or without a reconstructive procedure, aiming for microscopically clear margins (R0). The importance of obtaining clear margins has widely been stressed in the literature, but no consensus has been achieved in terms of distance of clearance. Some authors point out that the propensity of RIBAS satellite deposit formation indicates that 2–5 cm margins are preferable [18]. On the other hand, Cohen-Hallaleh classified resection as R0 if resection margins were >1 mm circumferentially. The same team reported that patients who developed local recurrence after RIBAS treatment had closer margins than those who did not (1 cm vs. 2.5 cm). However, margins were not found to be independent prognostic factor for oncological outcomes [15]. Surgical treatment of RIBAS patients in large-volume tertiary centers, rather specialized in breast and sarcoma treatment, increases the chances of obtaining negative margins, thus yielding better long-term outcomes in these patients [26]. Some studies suggest that the aim of surgical treatment should be excision of all irradiated tissue [18]. All patients in our study had resectable disease at diagnosis and underwent total mastectomy, with four of them needing pectoral defect reconstruction. R0 resection (>1 mm) was achieved in all cases, but margins were not routinely reported by pathologists, except in two patients (patients 8 and 9) who had 15–80 mm and 40–100 mm marginal clearance, respectively. 

In some centers, patients with localized, but non-resectable, disease were treated with neoadjuvant chemotherapy. Cohen-Hallaleh reported in their study that seven patients received neoadjuvant chemotherapy, with three of them achieving sufficient tumor downsizing to consider surgical resection [15]. Even though neoadjuvant chemotherapy may yield certain benefits to some patients, this approach has not been widely adopted. The decision regarding the appropriateness of neoadjuvant chemotherapy for each individual patient should be considered through multidisciplinary team meetings [19]. Combining surgical treatment with neoadjuvant/adjuvant therapy improves local disease control according to some authors, but not systemic control and overall survival [5]. Anthracycline monotherapy, or in combination with ifosfamide, represents the treatment of choice for metastatic soft tissue sarcomas (18). Based on multidisciplinary team decisions, four patients in our study received adjuvant chemotherapy. Three of them received doxorubicin, while one patient received doxorubicin and cyclophosphamide. The ANGIOTAX, phase II, French trial showed clinical benefit for patients with unresectable angiosarcomas treated with weekly Paclitaxel [27]. Another study reported the efficacy of Bevacizumab, a monoclonal antibody targeting VEGF, in the treatment of angiosarcoma [28]. These therapeutics were not available in our country for the given indication.

None of our patients received adjuvant RT for RIBAS. Taking into consideration the fact that RIBASs are radiation-induced tumors, as well as the total RT doses administered during BC treatment, the use of additional irradiation remains controversial. Rombouts reported no difference in survival between patients who received RT in combination with surgical treatment, compared to surgery alone (9). On the other hand, Depla suggested that surgery in combination with RT may provide benefits in local disease control [24]. Other studies also reported that re-irradiation, in combination with hyperthermia, with or without surgery, could also improve local control [29].

RIBASs are generally considered aggressive tumors with dismal prognosis. Even with the best treatment, local recurrence rates remain high. The estimated median LRFI in our study was 10 (95% CI >3) months, with a local recurrence rate (LRR) of 78%. Depla et al. reported, in large systematic review, the median LRFI for RIBAS patients treated only with surgery to be 12 months [24]. Other teams reported a median LRFI of 6 months with an LRR of 61% [30]. The patients in our study showed a median OS of 31 months, with a 3-year survival rate of 15.2% (95% CI 2.5–91.6%), which is worse compared to some other studies. Rombouts reported a 5-year OS of 40.5% and 10-year OS of 25.2%, while Depla reported a 5-year OS of 43% [9,24]. Another study showed a 2-year OS of 71.1% [15].

## 5. Conclusions

RIBAS is a rare but increasingly prevalent adverse event of RT, marked by an aggressive disease course and high relapse rates. So far, evidence in the literature, based on limited clinical experience, shows that three factors might be of critical importance for improving patients’ outcomes: the awareness of possibility for RIBAS to develop after breast/chest irradiation, the timely diagnosis with biopsy of any skin lesion that presents after breast/chest irradiation, and radical surgical excision with wide clear margins. More reports and multicentric studies on this rare entity can help enlighten the persistent problem of RIBAS management.

## Figures and Tables

**Figure 1 diagnostics-14-02326-f001:**
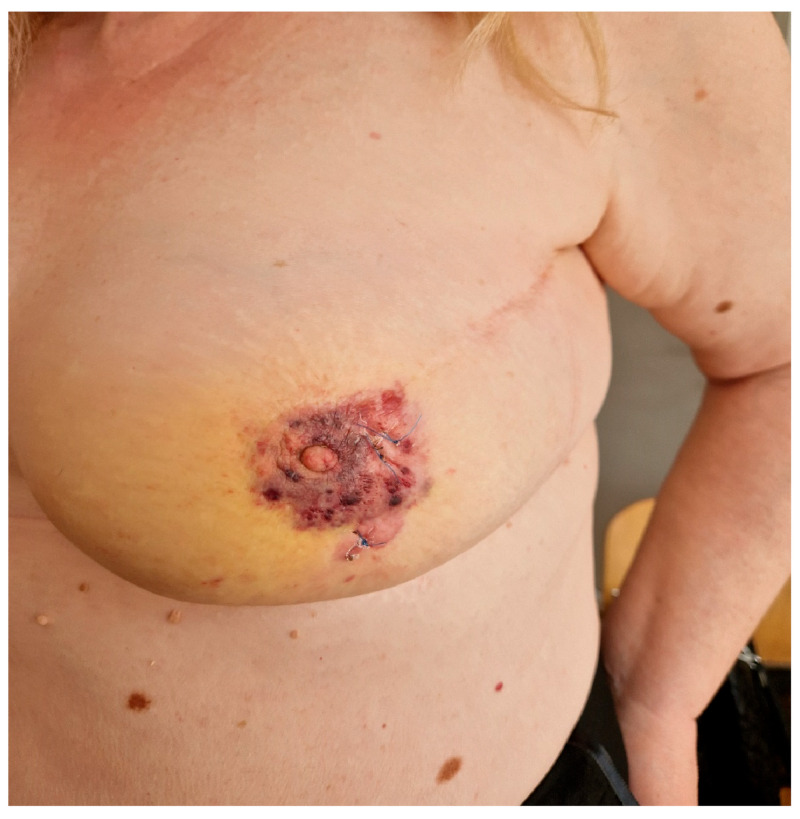
Clinical presentation of radiation-induced breast angiosarcoma in one of the patients included in this study.

**Figure 2 diagnostics-14-02326-f002:**
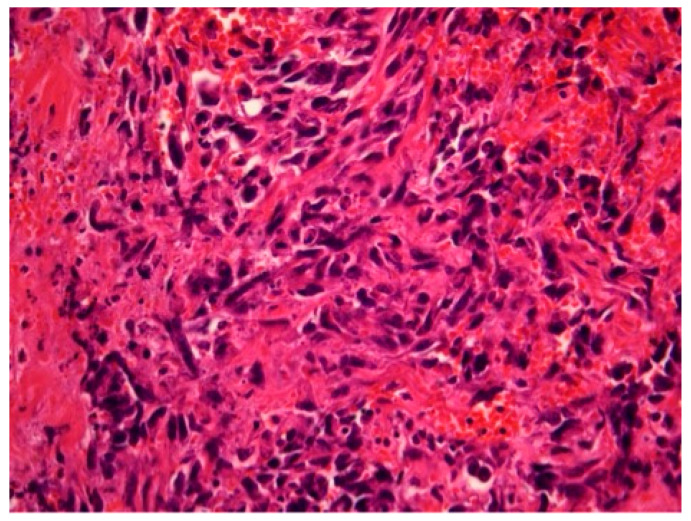
Radiation-induced breast angiosarcoma—microscopic analysis, HE staining, 40 × 1.

**Figure 3 diagnostics-14-02326-f003:**
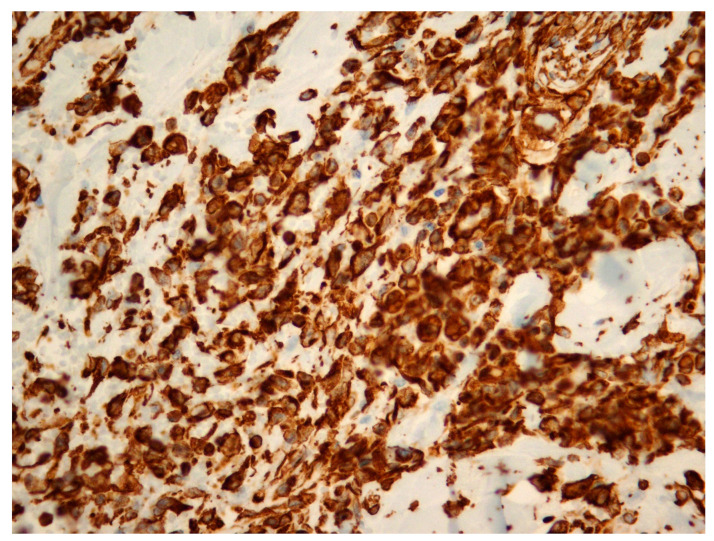
Radiation-induced breast angiosarcoma—microscopic analysis, Vimentin staining, 40 × 1.

**Figure 4 diagnostics-14-02326-f004:**
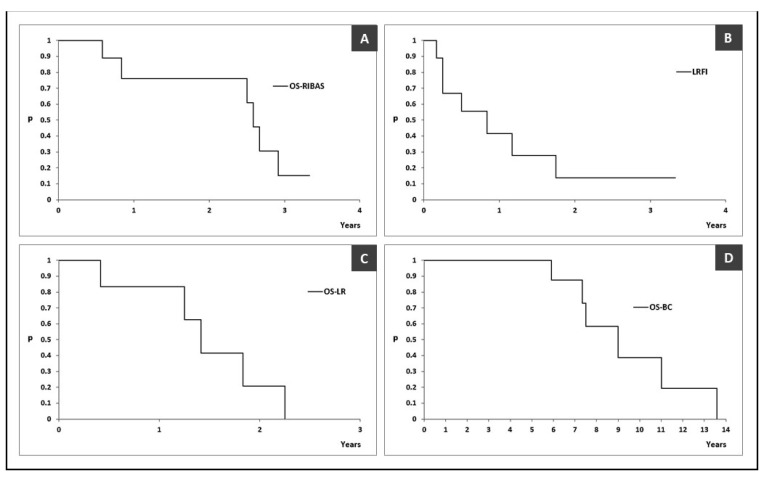
Kaplan–Meier curves reporting outcomes in patients with radiation-induced breast angiosarcoma. (**A**) overall survival of patients with radiation-induced breast angiosarcoma, defined as the time from initial RIBAS treatment to death or the last follow-up (OS-RIBAS); (**B**) local recurrence-free interval, defined as the time from initial RIBAS surgery to first pathohistologically proven local recurrence (LRFI); (**C**) overall survival after RIBAS local recurrence (OS-LR); (**D**) overall survival after RIBAS local recurrence (OS-BC). OS-RIBAS—overall survival of patients with radiation-induced breast angiosarcoma, defined as the time from initial RIBAS treatment to death or the last follow-up; LRFI—local recurrence-free interval, defined as the time from initial RIBAS surgery to first pathohistologically proven local recurrence; OS-LR—overall survival after RIBAS local recurrence; OS-BC—overall survival after initial breast cancer treatment.

**Table 1 diagnostics-14-02326-t001:** Initial breast cancer characteristics including patients’ age, disease stage, histology and molecular subtype, and treatment algorithm.

Pt	Age at BC Dg	BC Stage *	BC Histology	Surgery	Adj ChT **	Adj HT **	Adj RT **
1	53	T1N0M0	CDI, G2 ER+ PR+ HER2−	Lumpectomy + ALND	Yes	No	50 Gy/25 f + boost 10 Gy
2	64	T1N0M0	CDI, G2 ER+ PR+ HER2−	Lumpectomy + ALND	No	Yes	50 Gy/25 f
3	60	T2N0M0	other, G2 ER+ PR+ HER2−	Lumpectomy + ALND	No	Yes	50 Gy/25 f
4	52	T2N0M0	other, G1 ER+ PR+ HER2−	Lumpectomy + ALND	No	Yes	50 Gy/25 f + boost 10 Gy
5	31	T1N1M0	CDI, G2 ER+ PR+ HER2−	Lumpectomy + ALND	Yes	Yes	60 Gy/30 f
6	62	T1N0M0	CLI, G2 ER+ PR+ HER2−	Lumpectomy + SLNB	No	Yes	50 Gy/25 f
7	46	T1N0M0	other, G2 ER+ PR+ HER2−	Lumpectomy + ALND	No	Yes	50 Gy/25 f
8	51	T1N1M0	CLI, G2 ER+ PR+ HER2−	Lumpectomy + ALND	No	Yes	50 Gy/25 f + boost 10 Gy
9	64	T1N0M0	CDI, G2 ER+ PR+ HER2−	Lumpectomy + SLNB	No	Yes	42.4 Gy/16 f

Pt—patient; BC—breast cancer; Dg—diagnosis; Adj—adjuvant; ChT—chemotherapy; HT—hormonotherapy; RT—radiotherapy; CDI—ductal invasive cancer; CLI—lobular invasive cancer; other—non-CDI and non-CLI types (papillary intracystic cancer in patient 3; tubulo-lobular cancer in patient 4; tubular infiltrative cancer in patient 7); ER—estrogen receptor; PR—progesterone receptor; HER2—human epidermal growth factor receptor 2; ALND—axillary lymph node dissection; SLNB—sentinel lymph node biopsy; Gy—gray; f—fraction. * stage was defined based on the AJCC classification that was valid at the time of diagnosis. ** adjuvant treatment was administered based on national treatment guidelines for specific disease stages that align with ESMO recommendations.

**Table 2 diagnostics-14-02326-t002:** Radiation-induced breast angiosarcoma clinical presentation.

Patient	Latency Period * (Months)	RIBAS Size (mm)	RIBAS Grade
1	124	90	G3
2	55	43	G2
3	95	7	ND
4	58	60	ND
5	57	120	G2
6	55	40	G1
7	91	40	G2
8	69	55	G2
9	36	40	G2

RIBAS—radiation-induced breast angiosarcoma; ND—no data. * time from completed radiotherapy to RIBAS occurrence.

**Table 3 diagnostics-14-02326-t003:** Radiation-induced breast angiosarcoma treatment characteristics.

Pt	Surgery	Defect Reconstruction	Adj ChT *	Adj RT	Local Recurrence	LR Adj ChT *	Distant Metastases
1	TM	No	Doxo	No	Yes	No	No
2	TM	No	Doxo	No	Yes	Yes	No
3	TM	No	No	No	Yes	No	No
4	TM	LSF	No	No	Yes	No	Yes (lungs)
5	TM	Wolf	Doxo	No	No	No	No
6	TM	No	No	No	No	No	No
7	TM	LDF	AC	No	Yes	No	Yes (bone)
8	TM	FCF	No	No	Yes	Yes	Yes (bone)
9	TM	No	No	No	Yes	No	No

Pt—patient; Adj—adjuvant; ChT—chemotherapy; RT—radiotherapy; LR—local recurrence; TM—total mastectomy; LSF—local skin flap; Wolf—full-thickness skin graft; LDF—latissimus dorsi flap; FCF—fasciocutaneous flap; Doxo—doxorubicin as monotherapy; AC—doxorubicin and cyclophosphamide. * treatment was administered based on national treatment guidelines for specific disease stages that align with ESMO recommendations.

## Data Availability

Data included in the article.

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
