# Peer review of "Radiation-Induced Breast Angiosarcoma—A Single-Institution Experience"

_diagnostics, 2024, doi:10.3390/diagnostics14202326_

Round 1

Reviewer 1 Report

Comments and Suggestions for Authors

Well presented data about RIBAS. I'm thinking, if there was any observation regarding the years of the diagnosis of these cases, if the neoplasm is somehow related to the equipment used for the radiation. 

Author Response

Comment: Well presented data about RIBAS. I'm thinking, if there was any observation regarding the years of the diagnosis of these cases, if the neoplasm is somehow related to the equipment used for the radiation. 

Response: Dear, thank you for your comment. All the patients were treated on linear accelerators with 6MV energies (Elekta Synergy up to 2018, and then Varian Clinac and TrueBeam machines till 2023) , radiotherapy planing was conducted with 3d conformal radiotherapy technique for all patients. Our institute began to threat breast cancer patients on linear accelerators from 2008, so there was no significant differences in the time span regarding radiotherapy equipment. All patients were irradiated under the same technical conditions. We definitely do not have a large number of RIBAS compared to the number of irradiated patients.

We added those technical data regarding radiotherapy. Here you have attached updated file.

Best regards!

Reviewer 2 Report

Comments and Suggestions for Authors

1.     In the Abstract and Introduction, you mentioned 0,3%. I didn't understand the meaning of it. Is it 0.3%, 3%, or 0–3%? Please clarify.

2.     According to your study, you found 9 RIBAS patients out of 10,834 BC patients, which comes to approximately 0.08%. What is your interpretation of this incidence?

3.     There are a couple of studies (PMID: 31367294; PMID: 28794852) similar to yours. How do your findings differ from theirs?

Author Response

Comment 1: In the Abstract and Introduction, you mentioned 0,3%. I didn't understand the meaning of it. Is it 0.3%, 3%, or 0–3%? Please clarify.

Response: Dear Editor,

It is a technical mistake. It is 0.3% of all irradiated patients who developed RIBAS.

Comment 2: According to your study, you found 9 RIBAS patients out of 10,834 BC patients, which comes to approximately 0.08%. What is your interpretation of this incidence?

Response: We had 10,834 patients treated for breast cancer at our Institute. Not all of them received breast conserving surgery. In the section results in our paper, in the first sentence, row 83 and 84, we mentioned that 7647 patients received postoperative radiotherapy, meaning that these patients underwent breast-conserving surgery and met the criteria for this. Out of these 7647 patients, 9 developed RIBAS, which is approximately 0.12%, and is similar to other studies (PMID: 28794852).

Comment 2: There are a couple of studies (PMID: 31367294; PMID: 28794852) similar to yours. How do your findings differ from theirs?

In comparison with similar studies, there are generally no significant deiferences in the results. This disease is quite rare and large prospective studies are lacking to obtain more accurate and reliable results on larger samples. In our study, compared to the studies you mentioned, a slightly higher rate of local recurrence is noted, although all our patients after sarcoma surgery were R0. The reason for this could be that most patients had grade 2 tumors and higher.